# Testing the Motor Competence and Health-Related Variable Conceptual Model: A Path Analysis

**DOI:** 10.3390/jfmk3040061

**Published:** 2018-11-28

**Authors:** Ryan Donald Burns, You Fu

**Affiliations:** 1Department of Health, Kinesiology and Recreation, University of Utah, Salt Lake City, UT 84112, USA; 2School of Community Health Sciences, University of Nevada Reno; Reno, NV 89557, USA

**Keywords:** anthropometric measures, child motor development, exercise behavior, exercise motivation, health behaviors

## Abstract

The purpose of this study was to empirically test a comprehensive conceptual model linking gross motor skills, school day physical activity and health-related variables in a sample of sixth graders. Participants were a convenience sample of 84 sixth grade students (Mean age = 11.6 ± 0.6 years). Gross motor skills were assessed using the Test of Gross Motor Development-3rd Edition (TGMD-3), school day physical activity was assessed using pedometers, health-related fitness was assessed using Progressive Aerobic Cardiovascular Endurance Run (PACER) laps, perceived competence assessed using a validated questionnaire and the health-related outcome was assessed using Body Mass Index (BMI). The relationship between school day step counts and TGMD-3 scores was mediated through both perceived competence and PACER laps (*p* = 0.015) and the direct path coefficient between TGMD-3 scores and BMI was statistically significant (*b* = −0.22 kg/m^2^, *p* < 0.001). Overall there was good model fit with all indices meeting acceptable criteria (*χ*^2^ = 3.7, *p* = 0.293; Root Mean Square Error of Approximation (RMSEA) = 0.062, 90% Confidence Interval (C.I.): 0.00–0.23; Comparative Fit Index (CFI) = 0.98; Tucker-Lewis Index (TLI) = 0.96; Standardized Root Mean Square Residual (SRMR) = 0.052). The comprehensive conceptual model explaining the inter-relationships among motor competence and health-related variables was empirically validated with the relationship between physical activity and gross motor skills mediated through both perceived competence and cardiorespiratory endurance in a sample of sixth graders.

## 1. Introduction

Fundamental gross motor skills associate with children’s physical health, wellbeing and performance in activities of daily living [1,2]. Mechanisms for these links include that gross motor skills help children control their bodies, manipulate their environment and form complex skills involved in sports and other recreational activities, which may facilitate optimal growth and development [3]. Fundamental gross motor skills manifest from rudimentary phases of infancy to complicated movements that serve as building blocks for complex movements [4]. A growing body of research shows that these skills may facilitate participation in daily physical activity, which has its own health and wellness benefits [5]. Physical activity has been shown to be influenced by behavioral and psychological determinants across the lifespan [6,7]; gross motor skills may also correlate with these determinants (e.g., independent mobility and active transport). Unfortunately, fundamental motor skills do not develop naturally over time but rather need to be taught and reinforced to the developing child so that they can participate successfully in unstructured and structured physical activity [8].

Stodden et al. [9] proposed a conceptual framework linking improvements in gross motor skills with increases in physical activity in children and adolescents, which will further lead to decreases in cardio-metabolic disease risk. Some of these relationships have been empirically tested and may be partially mediated through health-related fitness [10,11,12,13,14]. The strength of these relationships depends on the current age-related stage of a child’s development [15], as the direction of inferred causation is moderated by whether a child is in early, middle, or late childhood [15]. Indeed, much research supports a link between gross motor skills and physical activity and there is evidence that this relationship is bidirectional in nature and is also mediated through the motivational variable of perceived competence [11,14,16].

After proposal of Stodden and colleagues’ model [9], research accumulated examining the inter-relationships among the proposed variables. Robinson et al. [17] compiled and elaborated on the mounted evidence in a narrative review linking gross motor skills, physical activity, health-related fitness and health outcomes. The pooled evidence yielded derivation of a similar conceptual model, linking physical activity to gross motor skills via perceived competence and health-related fitness, which in turn relates to a child’s weight status. Differences between the aforementioned conceptual models include which construct ultimately associates with a health indicator (e.g., weight status), physical activity or gross motor competence, however it is postulated in both models that these relationships are bidirectional in nature [17]. Research supports pieces of this conceptual model, however testing the model in its entirety has been precluded in the pediatric population. In their narrative review, Robinson and colleagues [17] recommended that researchers should explore the mediating effect of both perceived competence and health-related fitness in the relationship between motor competence and physical activity, explore discrete time periods for physical activity participation (e.g., school hours) and provide specificity in the direction of inferred causation among variables within the conceptual model.

No study to date has empirically tested the motor competence and health-related variable conceptual model in its entirety using a path analytic approach. Doing so will provide the relative strength of the aforementioned relationships, which can help in deriving interventions targeting these constructs. Also, the degree to which the link between gross motor skills and physical activity is mediated through both perceived competence and health-related fitness (e.g., cardiorespiratory endurance) has yet to be quantified and it is unknown whether if there is any residual relationship after accounting for the potential mediators. Even though many of the relationships within the proposed conceptual model are theoretically bidirectional, in the testing of the model in the current study, the physical trait of Body Mass Index (BMI) will be the health-related outcome and the behavior of school day physical activity will be the primary antecedent (predictor) within a fully recursive (unidirectional causative) model. This is because school- and community-based intervention-elicited behavior change is often utilized to positively influence physical traits (e.g., health-outcomes) over time and thus makes the most theoretical sense from the perspective of future intervention research. Testing a comprehensive conceptual model will provide further validation of the model and may spur testing of the models using larger samples of children or adolescents within a specific stage of physical development, which will provide information onto what constructs/variables to target during school and/or community-based interventions. Therefore, the purpose of this study was to test a comprehensive conceptual model linking gross motor skills, physical activity and health-related variables in a sample of sixth graders using path analysis. 

## 2. Materials and Methods

### 2.1. Participants

According to Kline et al. [18], the minimum adequate sample size for path analyses should be at least 10 times the number of parameters. There are 5 parameters in this study’s analysis. Participants were a convenience sample of 84 sixth grade students (Mean age = 11.6 ± 0.6 years; 40 girls, 44 boys) recruited one public Zoom school located in the Western U.S. Zoom schools are given supplemental funding for tutoring, smaller class sizes and extended learning opportunities and the majority of students within these schools have English as their second language. Zoom schools do not have organized physical education classes but have scheduled recess every day for 15 min after lunch. Exclusion criteria were any conditions that precluded students from participating in motor skill and health-related fitness testing. Approximately 67 participants were of Hispanic ethnicity (79.7%), 7 were African American (8.3%), 5 students were Caucasian (5.9%), 3 were Asian American (3.6%) and 2 self-reported as “Other” (2.4%). Participants were recruited from three classrooms. Participants provided written assent and parents or guardians provided written consent prior to data collection. The University of Nevada Reno Institutional Review Board approved the protocols employed in the study (project number: 1018396-4; 30 June 2017).

### 2.2. Physical Activity Assessment

Physical activity was the exogenous predictor variable with the conceptual model. Physical activity was assessed using school day step counts. School-day step counts were measured using Yamax DigiWalker CW600 pedometers (Yamax Corporation, Tokyo, Japan). Yamax DigiWalker models have been shown to provide an accurate recording of steps [19] and to be a reliable and valid measure of free-living physical activity [20]. The pedometers were worn for one school week (Monday–Friday) between the hours of 8 am and 3 pm. The pedometers were worn at waist level, in line with the right knee. Each pedometer had an identification number taped onto the device that was linked to each participant’s name. Data recorded within the spreadsheet included each participant’s daily pedometer step count, the average daily step count and specific daily wear time. The unit of analysis was average steps during school time across the 5 school days.

### 2.3. Perceived Competence Assessment

Perceived competence was a mediator variable within the conceptual model. Perceived Competence was assessed using the Perceived Competence Scale for Children (6 items) [21]. Perceived competence was on a 4 point scale consisting of example statements such as “I could not do better at physical activity” or “I am better at physical activity,” with the adolescents indicating on a continuum whether a respective statement is “Sort of true for me” to “Really true for me.” Cronbach’s alpha for the scale was determined to be acceptable (*α* = 0.89).

### 2.4. Gross motor Skill Assessment

Gross motor skills was an endogenous variable within the conceptual model. The Test for Gross Motor Development 3rd Edition (TGMD-3) was used to assess gross motor skills. Psychometric properties of the TGMD-3 have been reported with high levels of reliability and validity and is an updated test from the TGMD-2 [22,23]. The TGMD-3 assessed gross motor skills across thirteen movements. Movement skills were assessed across separate locomotor and ball skill subtests. The locomotor subtest items included running, skipping, sliding, galloping, hopping and horizontal jumping. The ball skill subtest items comprised of overhand throwing, underhand throwing, catching, dribbling, kicking, one-hand striking and two-hand striking. Each participant performed the test items across two trials scored based on the respective movement skill’s specific performance criteria (0 = did not perform correctly; 1 = performed correctly). The locomotor and ball skill subtest scores were 46 and 54 respectively and the total TGMD-3 scores were out of 100. Motor skill competency was scored using a total gross motor skill score. One member of the research team collected locomotor information at each school and one member of the research team collected ball skill information at each school to maintain testing consistency. Intra-observer and inter-observer reliability were tested on a sixth grade at a different class not recruited for the current study using both live and video scoring. The Intraclass Correlation Coefficient (ICC) = 0.91 for intra-observer agreement and ICC = 0.91 for inter-observer agreement.

### 2.5. Health-related Fitness Assessment

Two domains of health-related fitness were assessed: body composition and cardiorespiratory endurance. Body mass Index (BMI) was the endogenous outcome variable within the conceptual model. BMI was obtained by dividing weight in kilograms by the square of a participant’s height in meters. Height was measured to the nearest 1 cm using a stadiometer (Seca 213; Hanover, MD, USA) and weight was measured to the nearest 0.1 km using a medical scale (BD-590; Tokyo, Japan). Height and weight were collected in a separate room during physical education class. Cardiorespiratory endurance was assessed using the 20-meter Progressive Aerobic Cardiovascular Endurance Run (PACER) that was also administered during physical education class. PACER was a mediator variable within the conceptual model. PACER has been shown to be a reliable and valid test of cardiorespiratory endurance in children [24,25]. PACER was conducted on a marked gym floor with a compact disk providing background music. Each participant was instructed to run across a 20-meter distance within an allotted time frame. The allotted time given to reach the specified distance shortened incrementally as the PACER progressed. If a participant twice failed to reach the other floor marker, the test ended. The final score was recorded in PACER laps.

### 2.6. Statistical Analysis

Data were screened for outliers using box plots and z-scores (with a ±2.5 z cut-point) and checked for Gaussian distributions using k-density plots. Differences between sexes on all observed variables were examined using paired *t*-tests. Effect sizes were calculated using Cohen’s delta (d) with *d* < 0.20 indicating a small effect, *d* = 0.50 indicating a medium effect and *d* > 0.80 indicating a large effect [26]. A path analysis was employed using STATA’s “SEM Builder.” The lone exogenous variable (variable not affected by others) in the model was school day step counts. Endogenous variables (variables affected by others within the model) consisted of perceived competence, PACER laps, TGMD-3 scores and BMI. Direct relationships examined in the model was the relationship between school day steps and TGMD-3 scores and the relationship between TGMD-3 scores and BMI. Indirect relationships examined in the model included the relationship between school day step counts and TGMD-3 scores mediated through perceived competence and the relationship between school day step counts and TGMD-3 scores mediated through PACER laps. Indirect, direct and total effects were calculated using STATA’s “estat teffects” command. Reporting of the results involved communication of the unstandardized and standardized path coefficients. Acceptable overall model fit was indicated by a non-statistically significant chi-squared statistic [27], a Root Mean Square Error of Approximation (RMSEA) < 0.08 [28], a Comparative Fit Index (CFI) > 0.95 [29], a Tucker-Lewis Index (TLI) > 0.95 [27] and a Standardized Root Mean Square Residual (SRMR) < 0.08 [29]. Model fit at the equation-level was also assessed using the multiple correlation coefficient (R) and the coefficient of determination (R^2^). Alpha level was set at *p* < 0.05 and all analyses were conducted using STATA v.15.0 statistical software package (College Station, TX, USA).

## 3. Results

The descriptive statistics are presented in Table 1. Girls had higher BMI compared to boys (mean difference = 3.03 kg/m^2^, *p* = 0.023, *d* = 0.57). However, boys ran more PACER laps (mean difference = 13.1 laps, *p* = 0.014, *d* = 0.61) and had higher ball skill scores (mean difference = 8.5, *p* < 0.001, *d* = 1.20) and TGMD-3 total scores (mean difference = 11.7, *p* < 0.001, *d* = 1.11) compared to girls. There were no statistically significant differences between sexes on age (*p* = 0.719), school day steps (*p* = 0.062), perceived competence (*p* = 0.078), or locomotor skills (*p* = 0.061).

Figure 1 presents the results of the path analysis where both unstandardized and standardized path coefficients are communicated within the path diagram. All path coefficients were statistically significant, except for the direct relationship (direct effect) between school day step counts and TGMD-3 scores (*p* = 0.320). The relationship between school day step counts and TGMD-3 scores was mediated through both perceived competence and PACER laps (indirect effect = 0.001, 95% C.I.: 0.0002–0.0023, *p* = 0.015). Approximately 60% of the relationship between school day steps and TGMD-3 was mediated through perceived competence and PACER laps. The residual error variances between Perceived Competence and PACER Laps were correlated. There was overall good model fit with all indices meeting the acceptable criteria (χ^2^ = 3.7, *p* = 0.293; RMSEA = 0.062, 90% C.I.: 0.00–0.23; CFI = 0.98; TLI = 0.96; SRMR = 0.052). Table 2 presents the statistics related to equation-level goodness-of-fit, which provides information related to the effect size at the equation-level. The equation characterized with the lowest explanatory power was on the perceived competence outcome (7.3% of the variance explained). All other equations were characterized as having similar explanatory power on each respective outcome: TGMD-3 scores (23.3% of the variance explained), PACER laps (24.8% of the variance explained) and BMI (22.2% of the variance explained).

## 4. Discussion

The purpose of this study was to test a motor competence and health-related variable conceptual model on a sample of sixth grade students. The results indicated that the relationship between school day step counts and TGMD-3 scores was mediated through perceived competence and PACER laps and the direct relationship between TGMD-3 scores and BMI was statistically significant. The results from this analysis provide information for researchers and practitioners deriving school- and community-based interventions. The results also may direct avenues for additional research exploring other conceptual models. Interpretation of these specific findings are discussed further.

It has been well documented that perceived competence is an important motivational construct that mediates the relationship between physical activity and gross motor competence [9,16,30]. The findings of the current study soundly echo the existing literature. Indeed, the relationship between school day steps and TGMD-3 scores became non-significant without the mediating effect of perceived competence and cardiorespiratory endurance. According to Stodden et al. [9], as compared to younger children, children within the upper elementary and middle school age groups are characterized as having a development period during which perceived competence plays a more important role in mediating the relationship between physical activity and gross motor skills. As a result, it is expected that perceived competence will demonstrate a stronger mediating effect on motor skill competence and physical activity during late childhood than early childhood. This could be possibly illustrated by the mechanism of cognition and peer influence. Specifically, development of growing children’s cognitive capacity leads to the comparison between themselves to their peers during physical education class and consequently perceived competence more closely influences their actual motor skill competence [31]. Therefore, children with higher perceived competence are more active and more physically fit than those with low perceived competence [32,33]. In this study, school day physical activity explained 7.3% of the variance in children’s perceived competence and the coefficient was lower than a previous study by Davison, Downs and Birch [34], who reported that children’s perceived competence shared 27% of the variance with physical activity using a similar path analysis. Several plausible reasons may account for the differences, first of all, the participants in Davidson et al. [34] were girls, while the current study was conducted in both sexes in a co-educational setting. Additionally, the present study employed TGMD-3 and pedometers to objectively measure participants’ gross motor skill and physical activity, respectively. Davidson et al. [32] did not test participants’ actual motor skill competence and physical activity was assessed using self-reported method, which may manifest limitations in exploring the relationship, as Stodden et al. [9] has concluded that mediating effect of perceived competence on physical activity in middle to late childhood is largely based on the actual level of gross motor skills.

Stronger relationships within the tested model were observed between cardiorespiratory endurance (i.e., PACER laps) and TGMD-3 scores and between TGMD-3 scores and BMI. Cardiorespiratory endurance and body composition (using the BMI proxy) are two domains of health-related fitness. However, in both Stodden et al. [9] and Robinson et al. [17], weight status is an outcome within each conceptual model. The latter model postulates and the current study supports, that physical activity, cardiorespiratory endurance, gross motor skills and body composition are linked serially within a recursive path model. The relationship between physical activity and gross motor skills has been supported in the past [35], however this is the first study to establish that both perceived competence and cardiorespiratory endurance are mediators of effect. According to the observed results, higher levels of school day physical activity will increase cardiorespiratory endurance levels, which will further link to higher levels of gross motor skills and ultimately lower BMI. As stated previously, gross motor development does not naturally develop but needs to be taught and repetitively practiced and executed in order to be sufficiently developed [9]. Improving cardiorespiratory endurance may be a key component in this strategy. It may be that a sufficient level of fitness is needed for a child to be involved in activities that facilitate gross motor skill development. Children and adolescents with lower levels of cardiorespiratory endurance may not be able to participate in the activities needed to optimally develop gross motor skills or they may not spend long durations within activities that foster development of both locomotor and ball skills. Therefore, in order for a child to improve gross motor skills, higher levels of physical activity will need to both improve perceived competence and levels of cardiorespiratory endurance. Without this mediated path, there seems to be no relationship between physical activity and gross motor skill constructs in sixth graders, which emphasizes the importance of targeting perceived competence and cardiorespiratory endurance during specific school- or community-based interventions.

The final link within the conceptual model is the relationship between gross motor skills and BMI. This relationship was quite strong using the current study’s sample as 22% of the variance in BMI was explained by TGMD-3 scores. Like physical activity, the relationship between gross motor skills and weight status/BMI has been studied extensively. What is in question is the relative magnitude of the relationship during specific age-related periods of child development and the specific direction of inferred causation. D’Hondt et al. [36] found that weight status and gross motor skill scores were bidirectionally related within two separate mediation models, suggesting that children’s weight status negatively affects future gross motor skills and vice-versa. Drenowatz [37] suggests that a focus on motor competence can be used as a strategy for weight management in youth, stating that gross motor skills facilitate sustained physical activity, which over time will attenuate risk of overweight and obesity. Cheng et al. [38] found that higher BMI at age 5 predicted declines in gross motor competency from ages 5 to 10 years old. Although, it is suggested by Robinson and colleagues [17] that the relationship between these two constructs is indeed bidirectional, TGMD-3 scores and BMI were significantly related using the latter (BMI) as the outcome in the current sample of sixth graders. This relationship was direct with no testing of mediation or potential bidirectionality. Mechanisms for this significant relationship could be similar to those similar to that discussed in Drenowatz [37], although Barnett et al. [39] concluded that weight status had differential associations with aspects of gross motor competence. Barnett et al. [39], found that higher BMI was negatively correlated with motor coordination, stability and skill composite but not with object control after a systematic review of literature. This is in accordance with other work where variability in BMI is associated with aspects of gross motor skills characterized by movement in body mass (e.g., locomotor skills) [40,41]. Mediators of effect could play a role in this relationship and given the bidirectionally of the relationships within the conceptual model, it could be that physical activity and domains of health-related fitness again play a role within this potential causal pathway.

The results from this study yield avenues for future research. The relationship between school day physical activity and gross motor skills was mediated through perceived competence and PACER, therefore these constructs should be a target within physical activity programs that aim to improve gross motor skills. Studies have targeted perceived competence in the past when trying to elicit improvements in gross motor skills [11,39,42] however, there may be merit in targeting both perceived competence and cardiorespiratory endurance concomitantly. Because of the reciprocal relationship between gross motor skills and physical activity, targeting these mediators may exponentiate improvements in physical activity and gross motor skills within interventions because of a potential spiral for engagement (positive feedback) similar to that presented in Stodden and colleagues’ [9] model. In Robinson and colleagues’ [17] conceptual model, health-related fitness is identified as broad construct, not a specific domain. Other domains of health-related fitness, such as muscular strength and endurance or flexibility, may provide additional useful information. Also, in Robinson’s and colleagues’ conceptual model, there is a bidirectional relationship among many variables, therefore, there is merit testing a non-recursive path model using a larger and more diverse sample of children and adolescents, specifically testing reciprocal paths between physical activity and gross motor skills and between gross motor skills and BMI. Finally, as also recommended by Robinson et al. [17], testing the conceptual model during other time periods may provide novel information. School time is only a fraction of the day for many children and adolescents and is constrained by scheduling within the academic classroom; therefore, testing the relationships within the model outside of school and/or on the weekends may provide important information that can be used for future interventions.

There are limitations to this study that must be considered before the results can be generalized. First, the study sample consisted of students recruited from one school from the western U.S. characterized by distinct ethnic/racial representation; therefore, the results may not generalize to other geographical regions or to samples with different ethnic/racial characteristics. Second, the study design was cross-sectional; therefore, no causal inferences can be made. Third, only school day physical activity was assessed. The results may have differed if step counts were measured across the entire day. Fourth, the physical activity construct was assessed using pedometers. The construct validity may have been stronger if accelerometers were used to assess physical activity. Fifth, health-related fitness (cardiorespiratory endurance) was assessed using a field test. The construct validity of the health-related fitness construct may have been stronger if aerobic capacity was more directly measured in lab settings (i.e., measured VO_2 Peak_). Finally, sample size precluded testing a non-recursive model with reciprocal (bidirectional) relationships.

## 5. Conclusions

In conclusion, the motor competence and health-related variable model proposed by Robinson and colleagues [17] was validated in a sample of sixth graders. The model displayed acceptable fit and most path coefficients were significant within the tested model. The relationship between school day physical activity and gross motor skills was mediated through perceived competence and cardiorespiratory endurance; no significant direct relationship between the two constructs was observed. Results of the analysis suggest that the mediators between physical activity and gross motor skills may play a significant role in gross motor development. Ultimately, sixth graders with higher levels of gross motor skills had lower BMI, which may improve health risk if tracked through adolescence and into adulthood. Future research should validate the proposed model in larger and more diverse samples of children and adolescents using non-recursive models and also consider individual variability. Additionally, testing other domains of health-related fitness (e.g., muscular strength and endurance) may have merit and provide additional important information within the conceptual model. The conceptual model proposed by Robinson and colleagues [17] was validated in a sample of sixth graders and supports the important link among physical activity, gross motor outcomes and health in the pediatric population.

## Figures and Tables

**Figure 1 jfmk-03-00061-f001:**
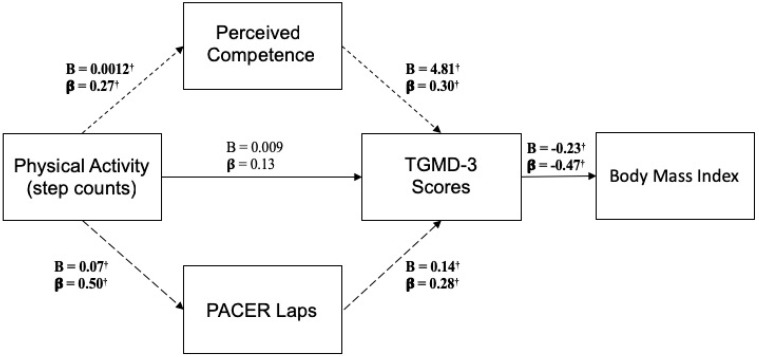
Path diagram aligning with the motor competence and health-related variable conceptual framework. *Note*: PACER stands for the Progressive Aerobic Cardiovascular Endurance Run; TGMD-3 stands for the Test of Gross Motor Development-3rd Edition; B denotes unstandardized coefficients; β denotes standardized coefficients; dashed lines represent a mediated (indirect) relationship; bold lines represent a non-mediated (direct) relationship; bold and † denotes statistical significance, *p* < 0.05.

**Table 1 jfmk-03-00061-t001:** Descriptive statistics (means and standard deviations).

	Girls (*n* = 40)	Boys (*n* = 44)	Total Sample (*n* = 84)
BMI (kg/m^2^)	23.7 ^†^ (5.9)	21.1 (4.3)	22.7 (5.5)
School Steps	3376 (1579)	4214 (1435)	3681 (1577)
PACER Laps	22.0 (19.4)	**35.3 ^†^ (20.9)**	26.4 (19.3)
Perceived Competence	3.0 (0.7)	3.4 (0.6)	3.3 (0.7)
Locomotor Skills	36.1 (5.8)	39.0 (6.3)	37.9 (6.1)
Ball Skills	39.4 (5.2)	**48.9 ^†^ (6.2)**	41.2 (7.2)
TGMD-3 Total Score	74.9 (7.8)	**88.1 ^†^ (10.9)**	80.5 (10.9)

BMI stands for Body Mass Index; PACER stands for the Progressive Aerobic Cardiovascular Endurance Run; TGMD-3 stands for the Test of Gross Motor Development-3rd Edition; bold and † denotes statistical differences between sexes, *p* < 0.05.

**Table 2 jfmk-03-00061-t002:** Equation-level goodness-of-fit-statistics.

Outcome Variables	Fitted Variance	Predicted Variance	Residual Variance	*R*	*R* ^2^
Perceived Competence	0.481	0.035	0.446	0.271	0.073
TGMD-3 Scores	119.722	27.880	91.841	0.483	0.233
PACER Laps	457.200	113.483	343.717	0.498	0.248
BMI (kg/m^2^)	28.103	6.247	21.857	0.471	0.222

TGMD-3 stands for the Test of Gross Motor Development-3rd Edition; PACER stands for the Progressive Aerobic Cardiovascular Endurance Run; BMI stands for Body Mass Index; *R* is the multiple correlation coefficient; *R*^2^ is the coefficient of determination.

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
