# Peer review of "Testing the Motor Competence and Health-Related Variable Conceptual Model: A Path Analysis"

_jfmk, 2018, doi:10.3390/jfmk3040061_

Round 1

Reviewer 1 Report

This is an interesting paper investigating the relationship between motor competence (TDMG-3) and health-related variables. All sections related to a scientific article fairly well presented and organized.

The introduction clearly reports the rational and the aim of the study. The method section is adequately presented (i.e., participants, physical activity assessment, perceveid competence assessment, gross motor skills assessment, health-related fitness assessment and statistical analysis). The results section is fairly well organized and the related figure (Fig1) and tables (1 and 2) are not redundant with the text.The discussion is clearly articulated and the results discussed. The authors reports at the end of the discussion the limitations of their study.

I have only minor concerns that needs to be addressed to the authors;

1) Is there any inclusion or exclusion criteria? This should be clearly stipulated.

2) Please, the pedometer used should be adequately documented: Yamax Digiwalker CW-600 pedometer (Yamax Corporation, Tokyo, Japan).

Author Response

Reviewer 1:

-Is there any inclusion or exclusion criteria? This should be clearly stipulated.

-Thank you. Exclusion criteria were any condition that precluded students from participating in motor skill and health-related fitness testing (lines 97-98).

-Please, the pedometer used should be adequately documented: Yamax Digiwalker CW-600 pedometer (Yamax Corporation, Tokyo, Japan).

-Thank you, this information has now been provided (line 107).

Reviewer 2 Report

Although I think the abstract is clear and quite informative, the conclusions (lines 20-22) could be improved.

I think the Introduction could benefit from some of the findings of the papers published in JFMK in special issue on “Health Promotion in Children and Adolescents through Sport and Physical Activities” as well as from recent umbrella reviews (Cortis et al., 2017 Plos One; Condello et al., 2017 International Journal of Behavioral Nutrition and Physical Activity). Moreover, when testing the conceptual model, terms such as physical activity, determinants, mediators, moderators, should be clearly defined as they might lead to different interpretations. As data were collected during school days, I think information about the physical education time should be provided as this could have an impact on the results (how many hours per day or per week is the rule in that school? This could vary from school to school in the same country as well as from country to country).

I will suggest using standard deviation as ± instead of +/-

What about the use of pedometer and PACER in children? Is there any literature supporting their validity and reliability in children? I think this could be helpful, especially because reliability information has been provided for other variables.

Line 111. Is “the” a typo or something is missing in the sentence?

I think the end of the arrows in Figure 1 should be bigger so that one can immediately understand which way the direction is.

Author Response

Reviewer 2:

-Although I think the abstract is clear and quite informative, the conclusions (lines 20-22) could be improved.

-Thank you, the abstract conclusion was expanded upon (lines 22-26).

-I think the Introduction could benefit from some of the findings of the papers published in JFMK in special issue on “Health Promotion in Children and Adolescents through Sport and Physical Activities” as well as from recent umbrella reviews (Cortis et al., 2017 Plos One; Condello et al., 2017 International Journal of Behavioral Nutrition and Physical Activity). 

-Thank you, these suggested references have been provided in the Introduction section (lines 39-41, 361-368).

-Moreover, when testing the conceptual model, terms such as physical activity, determinants, mediators, moderators, should be clearly defined as they might lead to different interpretations. As data were collected during school days, I think information about the physical education time should be provided as this could have an impact on the results (how many hours per day or per week is the rule in that school? This could vary from school to school in the same country as well as from country to country).

-Thank you for this comment. Each variable was identified as either a predictor, outcome, exogenous, endogenous, and/or mediator variable within the conceptual model (lines 105, 116, 123-124, 142, 148-150).  School physical education characteristics are now given within the Participants section (lines 95-96).

-I will suggest using standard deviation as ± instead of +/-

-Thank you, this symbol has been replaced (lines 92, 156).

-What about the use of pedometer and PACER in children? Is there any literature supporting their validity and reliability in children? I think this could be helpful, especially because reliability information has been provided for other variables.

-Thank you, this information has now been provided in the respective Methods sections (lines 108, 149-150).

-Line 111. Is “the” a typo or something is missing                 in the sentence?

-Thank you, this sentence has been revised (lines 123-124).

-I think the end of the arrows in Figure 1 should be bigger so that one can immediately understand which way the direction is.

-Thank you, the end arrows’ size has been enlarged.